# Acute Respiratory Tract Infection and 25-Hydroxyvitamin D Concentration: A Systematic Review and Meta-Analysis

**DOI:** 10.3390/ijerph16173020

**Published:** 2019-08-21

**Authors:** Hai Pham, Aninda Rahman, Azam Majidi, Mary Waterhouse, Rachel E. Neale

**Affiliations:** 1Population Health Department, QIMR Berghofer Medical Research Institute, Herston, QL 4006, Australia; 2School of Public Health, The University of Queensland, Herston, QL 4006, Australia

**Keywords:** respiratory infection, vitamin D, systematic review, observational studies, 25-hydroxyvitamin D, meta-analysis, acute infection

## Abstract

Observational studies and randomised controlled studies suggest that vitamin D plays a role in the prevention of acute respiratory tract infection (ARTI); however, findings are inconsistent and the optimal serum 25-hydroxyvitamin D (25(OH)D) concentration remains unclear. To review the link between 25(OH)D concentration and ARTI, we searched PubMed and EMBASE databases to identify observational studies reporting the association between 25(OH)D concentration and risk or severity of ARTI. We used random-effects meta-analysis to pool findings across studies. Twenty-four studies were included in the review, 14 were included in the meta-analysis of ARTI risk and five in the meta-analysis of severity. Serum 25(OH)D concentration was inversely associated with risk and severity of ARTI; pooled odds ratios (95% confidence interval) were 1.83 (1.42–2.37) and 2.46 (1.65–3.66), respectively, comparing the lowest with the highest 25(OH)D category. For each 10 nmol/L decrease in 25(OH)D concentration, the odds of ARTI increased by 1.02 (0.97–1.07). This was a non-linear trend, with the sharpest increase in risk of ARTI occurring at 25(OH)D concentration < 37.5 nmol/L. In conclusion, there is an inverse non-linear association between 25(OH)D concentration and ARTI.

## 1. Introduction

Acute respiratory tract infection (ARTI) is very common, with most people experiencing at least one episode of ARTI each year [1]. ARTI includes upper respiratory tract infection (URTI) and lower respiratory tract infection (LRTI). The common cold and influenza are the most common ARTIs globally; the highest rate occurs during the winter months in temperate areas with little seasonal change in tropical regions [2]. During epidemic months, influenza can affect 20% to 50% of people worldwide [2]. Bronchitis and pneumonia are the two most common infections of the lower airways, and mortality due to LRTI is high [3]. In 2015, more than 2.8 million deaths worldwide were due to LRTI; children and the elderly are the most affected groups [3].

Vitamin D is critical for skeletal health and may play a role in other health outcomes, including infection. Vitamin D status is estimated by measuring the serum concentration of 25-hydroxyvitamin D (25(OH)D); this varies seasonally and the lowest concentration in the winter/spring months coincides with the highest ARTI incidence, suggesting a link between vitamin D and ARTI [4]. This is supported by laboratory studies demonstrating an important role of vitamin D in the immune system. Vitamin D promotes the elimination of pathogens and suppresses prolonged inflammatory responses [5,6,7]. It enhances the production of antimicrobial peptides such as defensins and cathelicidins, which offer natural protection against microbial pathogens [8].

Many studies have investigated the link between 25(OH)D serum concentration and ARTI and the effect of vitamin D supplementation on ARTI. Two recent meta-analyses of observational studies considered associations between 25(OH)D concentration and ARTI incidence in children [9,10]. One included case-control studies in children aged ≤ 5 years and found higher odds of 25(OH)D deficiency in those with LRTI [9]. The other found an inverse association between prenatal maternal 25(OH)D concentration and risk of ARTI in the offspring [10]. The most recent systematic review and meta-analysis including older children and adults was published in 2015 as a conference abstract only—it included 19 observational studies and found a significant inverse association between 25(OH)D concentration and risk of ARTI [11]. A systematic review of 25 observational studies and 14 randomised-controlled trials (RCTs) was completed in 2013 [12]. It concluded that there was an inverse association between 25(OH)D concentration and risk of ARTI, but did not include a meta-analysis [12]. Some observational studies have found an inverse link between 25(OH)D concentration and severity of ARTI, as measured by duration of the illness, hospitalisation and severity index [13,14,15,16]; however, there has been no meta-analysis of these findings.

Results from RCTs investigating the effect of vitamin D supplementation on ARTI are inconsistent. Two meta-analyses found a significant benefit of vitamin D supplementation on ARTI [17,18] while another three did not [19,20,21]. There are indications of a greater protective effect in people with marked vitamin D deficiency [20,22], but the trials were unable to indicate an optimal concentration of 25(OH)D. We therefore conducted a systematic review and meta-analysis of observational studies to evaluate the link between serum 25(OH)D concentration and the risk and severity of ARTI in adolescents and adults. Findings from this meta-analysis will provide insights into the influence of vitamin D on ARTI risk and severity, and give an indication of the optimal 25(OH)D concentration for ARTI prevention and management.

## 2. Materials and Methods

The study protocol was registered with the PROSPERO International Prospective Register of Systematic Reviews prior to commencement [23].

### 2.1. Search and Screening Strategy

PubMed and EMBASE databases were searched from their inception until 12th June 2019. Keywords were chosen from the Medical Subject Headings (MeSH) terms in PubMed and explosion (exp) of EMTREE terms in EMBASE. Essentially, we searched for the terms “vitamin d” or “25-hydroxyvitamin D” or “25OHD” or “25(OH)D” or “hypovitaminosis D” in combination with “respiratory tract infection*” or “respiratory infectio*” or “respiratory diseas*” or “pneumonia” or “influenza” or “bronchiolitis” or “common cold”. The complete search strategies are shown in the Appendix A.

All titles, abstracts and full text were independently screened by two investigators, Hai Pham and Aninda Rahman. Any discrepancies were resolved through consultation with Rachel Neale and Mary Waterhouse.

### 2.2. Inclusion/Exclusion Criteria

*Inclusion criteria:* The eligibility criteria were: (1) observational studies including cohort, case-control and cross-sectional studies; (2) published in English; (3) full text available; (4) reported the association between circulating 25(OH)D concentration and ARTI risk or severity; (5) the target population was healthy adults or adolescents aged 12 years and older.

*Exclusion criteria*: Studies investigating the link between 25(OH)D concentration and tuberculosis or chronic lung conditions such as chronic obstructive pulmonary disease and asthma were excluded.

### 2.3. Definition of Outcome

*Primary outcome*: The primary outcome was the risk of ARTI, defined as an acute infection of the respiratory tract in either the lower or upper airway or with the location not specified. ARTI was either self-reported via surveys or symptom diaries, or clinically confirmed with or without evidence from X-rays or laboratory tests.

*Secondary outcome:* The secondary outcome was the severity of ARTI, defined according to the duration of the illness, hospitalisation, admission to an intensive care unit, symptom severity score or index, or mortality.

### 2.4. Quality Assessment

We used the customised Newcastle-Ottawa scale (NOS) to assess the quality of each study [24]. The NOS tools are slightly different for each study design but generally include three main categories, namely: (1) selection of participants; (2) control for confounders; and (3) measurement of exposure or outcome (Appendix A).

### 2.5. Statistical Analysis Methods

We used STATA 13 (StataCorp, College Station, Texas, USA) and SAS 9.4 (SAS Institute Inc., Cary, NC, USA) for statistical analyses. We included studies in the meta-analysis that reported either:
A measure of association between 25(OH)D concentration and ARTI risk or severity (odds ratio (OR), relative risk (RR), hazard ratio (HR), mean difference (MD)) and their 95% confidence interval (CI) or standard deviation (SD); orSufficient data to derive two by two tables of ARTI risk, comparing the lowest versus the highest 25(OH)D category.


We included the estimate from the most fully adjusted model for each study in the meta-analysis comparing the risk of ARTI in the lowest versus highest 25(OH)D category. When a study reported results using different 25(OH)D thresholds, we used 50 nmol/L as it was the most commonly used threshold. Two studies reported HRs as the measure of association between 25(OH)D concentration and risk of ARTI [25,26]; we included these two studies in the meta-analysis, considering the HR as an approximation of the OR. A random-effects model was used to pool the results.

We used a method described by Greenland and Longnecker to estimate trends across categories of exposure to calculate the effect of each 10 nmol/L decrease in 25(OH)D concentration on the risk of ARTI [27,28]. This method estimates the covariances between multivariable-adjusted odds ratios using the number of cases in each exposure category. We used variance least-squares regression to compute the trends in two studies in which the number of cases in each exposure category was not reported [25,29]. The representative value for each 25(OH)D category was assigned using either the midpoint of the range or by subtracting or adding the half width of the adjacent exposure category for open-ended categories [30]. We also used fixed effects to fit a restricted cubic spline model with 3 knots to evaluate a potential non-linear dose-response association between 25(OH)D concentration and ARTI risk [28].

For the overall analysis of severity, we defined a severe ARTI to be one that had a moderate-to-high severity score or resulted in death (no studies based on other severity outcomes were included). We also performed a separate meta-analysis to assess the association between 25(OH)D concentration (lowest versus highest category) and ARTI mortality.

### 2.6. Assessment of Heterogeneity and Publication Bias

Heterogeneity across studies was assessed using I^2^ statistics. We conducted meta-analyses comparing the risk of ARTI in the highest versus the lowest 25(OH)D category stratified by: (i) infection location if reported (URTI or LRTI); (ii) outcome measurement methods (self-reported or clinically confirmed); (iii) mean 25(OH)D concentration in the study population (<60 nmol/L or ≥60 nmol/L; cut point chosen to maximize the number of studies included in each subgroup); and (iv) factors considered as confounders (fully adjusted model or crude/non-fully adjusted model) to explore potential sources of heterogeneity. We used funnel plots and the Egger test to test for publication bias.

## 3. Results

### 3.1. Identification and Selection of Studies

Figure 1 illustrates the study selection process. The PubMed and EMBASE searches yielded a total of 1589 records after removing duplicates. We screened the full text of 33 studies and excluded nine that did not meet the selection criteria. Of the 24 studies included, 10 reported the association between 25(OH)D and the risk of ARTI [25,29,31,32,33,34,35,36,37,38], eight reported on the severity of ARTI [14,16,39,40,41,42,43,44], and six reported on both the risk and severity [13,15,26,45,46,47].

### 3.2. Study Characteristics

The characteristics of the included studies are summarised in Table 1 and Table 2. There were three case-control studies, 13 cross-sectional studies and eight prospective cohort studies. The relationship between ARTI risk or severity and 25(OH)D status was reported either by categories of exposure or per unit increase in exposure. The cut point for 25(OH)D categories varied across studies; the upper limit of the lowest category ranged from 25 to 95 nmol/L, and the cut point for the highest category ranged from 25 to 120 nmol/L (Table 1 and Table 2). Serum 25(OH)D concentration was measured using liquid chromatography-mass spectrometry (LCMS) in four studies and non-chromatography assays in 19 studies. One did not specify which method was used to measure 25(OH)D concentration [44].

ARTI risk was measured by (i) self-report via survey questionnaire or symptom diaries (n = 10), or (ii) clinical diagnosis with or without confirmation from X-ray or a laboratory test (*n* = 6). ARTI severity was measured using a severity index (*n* = 6) such as CURB or CURB-65 (confusion, uraemia, respiratory rate, low blood pressure, age ≥ 65), pulmonary severity score (PSI), or symptom severity score (SSC); duration of the illness (*n* = 10); admission to hospital or intensive care units (*n* = 4); or mortality (*n* = 6). All 24 studies had a moderate-to-high quality score (Appendix A).

### 3.3. Association between 25(OH)D Concentration and Risk of ARTI

We included 14 of the 16 studies that assessed the association between 25(OH)D concentration and risk of ARTI in the meta-analysis comparing the risk of ARTI in the lowest versus the highest 25(OH)D category (*N* participants = 78,127) and 10 studies were included in the trend analysis (*N* participants = 69,048). Five studies reported risk of ARTI in at least three categories of 25(OH)D concentration and the number of cases by exposure category; these were included in the evaluation of whether there is a non-linear relationship between ARTI risk and 25(OH)D concentration (*N* participants = 37,902).

There was a significantly higher risk of ARTI in the lowest compared with the highest 25(OH)D category (pooled OR 1.83; 95% CI 1.42–2.37; I^2^ = 78.8%; *p* < 0.001) (Figure 2). The pooled OR per 10 nmol/L decrease in 25(OH)D was 1.02 (95% CI 0.97–1.07; I^2^ = 72.7%; *p* < 0.001) (Figure 3). There was a significant non-linear relationship (*p* for non-linearity = 0.029) with inflexion points at 60 nmol/L and 37.5 nmol/L. The steepest increased risk occurred below 37.5 nmol/L (Figure 4). Although both linear and spline models were significant, the spline model fitted the data better.

### 3.4. Association between 25(OH)D Concentration and Severity of ARTI

Five studies (*N* participants = 1495) were included in the meta-analysis of the odds of severe ARTI or mortality combined, comparing the highest and the lowest 25(OH)D category and four studies (*N* participants = 1422) were included in the mortality meta-analysis. The pooled ORs for severity/mortality combined and mortality separately were 2.46 (95% CI 1.65–3.66; I^2^ = 49.8%; *p* = 0.093) and 3.00 (95% CI 1.89–4.78; I^2^ = 66.7%; *p* = 0.029), respectively (Figure 5). The duration of ARTI was also inversely associated with 25(OH)D concentration; low 25(OH)D concentration was associated with a more prolonged ARTI in 7 out of 10 studies (Table 3).

### 3.5. Studies Excluded from the Meta-Analysis

Two studies were excluded from the meta-analysis of ARTI risk, and nine studies were excluded from the meta-analysis of ARTI severity. Reasons for exclusion and the main findings of the excluded studies are provided in Appendix A. Studies that reported mean (SD) or median (interquartile range (IQR)) of 25(OH)D concentration in people with severe versus non-severe ARTI or mean (SD) severity score by 25(OH)D category were excluded. As different scales were used to measure ARTI severity across studies, we could not pool the mean difference in severity score between the highest and the lowest 25(OH)D category. Overall, results from 6/9 excluded studies showed an inverse association between 25(OH)D concentration and severity of ARTI measured by duration of the illness, severity score, or the number of days absent from duty (Appendix A).

### 3.6. Subgroup Analysis and Publication Bias

We found evidence of significant heterogeneity (I^2^ = 78.8%, *p* < 0.001); therefore, we conducted stratified meta-analyses to explore potential sources of heterogeneity (Figure 6A–D). The association between 25(OH)D concentration and LRTI was stronger than for URTI, but the former had significant heterogeneity (Figure 6A). Similarly, the association was stronger for studies with mean 25(OH)D concentration < 60 nmol/L than in those with mean 25(OH)D concentration ≥ 60 nmol/L, but again with significant heterogeneity (Figure 6C). There was only a small difference in the pooled OR between studies that used a fully adjusted model and those that reported estimates from crude models or those with less complete adjustment, but the latter had significant heterogeneity (I^2^ = 87.2%, *p* < 0.001) (Figure 6D).

The funnel plot shows evidence of significant publication bias (*p* = 0.024) (Figure 7), suggesting that studies with small sample size and insignificant findings were not published.

## 4. Discussion

In this systematic review and meta-analysis, we observed significant associations between 25(OH)D concentration and both risk and severity of ARTI, but with significant heterogeneity and evidence of publication bias. There was a non-linear association between 25(OH)D concentration and risk of ARTI, with evidence of a more marked increased risk for 25(OH)D concentration below 37.5 nmol/L.

Findings from our meta-analysis of ARTI risk, which includes the largest number of participants, are consistent with results from previous reports. A meta-analysis published in 2015 as an abstract (*n* = 44,301) reported an increased risk of ARTI in those with 25(OH)D < 50 nmol/l compared with those ≥ 50 nmol/L. The odds ratio was higher than we found (2.63 vs. 1.83); their analysis included children as well as adults and they did not present an estimate for adults and adolescents only [11]. A systematic review including 25 observational studies (10 in adults) reported significant inverse association between 25(OH)D concentration and risk of ARTI, but no formal meta-analysis was conducted [12]. The link between 25(OH)D concentration and risk of ARTI was also reported separately for children and LRTI; a meta-analysis showed a higher prevalence of vitamin D deficiency (< 50 nmol/L) in children with LRTI (*n* = 550) [9].

This is the first study to our knowledge which performed a meta-analysis of the association between 25(OH)D concentration and severity of ARTI. We found a stronger association between 25(OH)D concentration and severity of ARTI than with risk of ARTI (OR 2.46 vs. 1.83). The result accords with findings from a systematic review that highlighted the potential link between vitamin D deficiency and severe LRTI in children, but no meta-analysis was included [9]. It is difficult to meta-analyse the association between 25(OH)D concentration and severity of ARTI because of variability across studies with respect to variability in assay methods used to measure serum 25(OH)D; the cut points used to categorise 25(OH)D concentration; the scales used to measure severity; and the measures used to estimate effect. We could only include five studies in the severity meta-analysis so were unable to assess the association within subgroups or investigate a potential non-linear trend.

Results from RCTs are inconsistent. Some trials found that vitamin D supplementation reduced the risk of ARTI [48,49,50] while others did not [51,52,53,54]. The inconsistency may be due to differences in study design, vitamin D supplement doses and regimens, and different baseline 25(OH)D concentration. The most recent meta-analysis, using individual participant data, found that vitamin D supplementation reduced the risk of ARTI more strongly in people with 25(OH)D concentration < 25 nmol/L [22]. Our findings suggest that supplementation may be of most benefit in people with a 25(OH)D concentration ≤ 37.5 nmol/L, with some benefit up to 60 nmol/L, although these values should be considered with caution in light of the different assays and markedly varying cut point used across studies.

We observed high heterogeneity, and this mostly persisted in subgroup analyses. Despite this, the direction of effect was consistent; lower 25(OH)D concentration was associated with increased risk or severity of ARTI. There are indications of publication bias, indicating that small studies that showed no significant association were either not identified or not published. Publication bias has also been observed in meta-analyses of RCTs [17,22]. It is thus possible that the benefits of vitamin D for reducing risk or severity of ARTI have been over-estimated.

## 5. Conclusions

Our study is the largest to date and to the best of our knowledge, is the first to include a meta-analysis of the association between 25(OH)D concentration and severity of ARTI. Our findings suggest an important role of vitamin D in prevention of ARTI risk and severity, particularly in people with low 25(OH)D concentration. However, it is challenging to identify an optimal 25(OH)D concentration or a concentration below which supplementation would be of benefit due to the lack of consistency in both 25(OH)D assays and reporting across studies. It is important to improve consistency of reporting, as well as assays, to enable the field to move forward.

## Figures and Tables

**Figure 1 ijerph-16-03020-f001:**
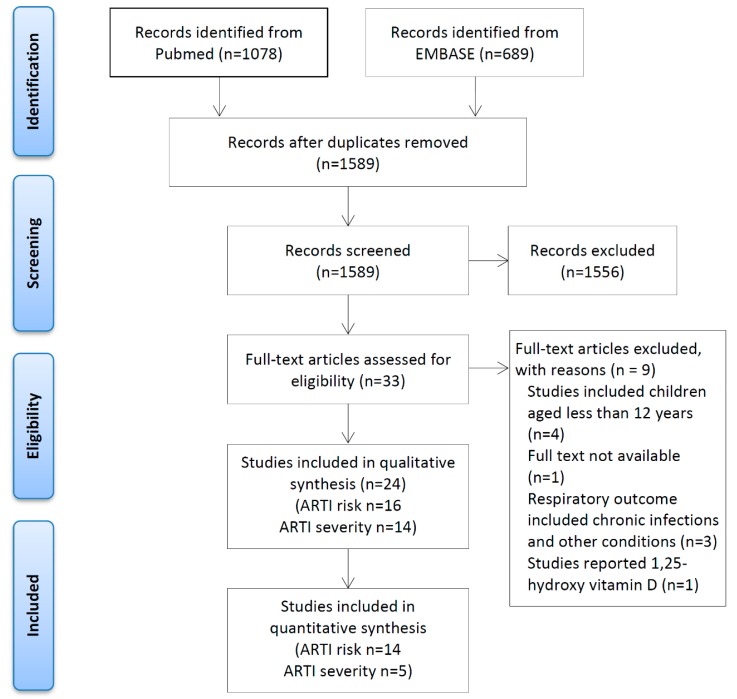
The Prisma flowchart for study selection process. Note: ARTI = acute respiratory tract infection.

**Figure 2 ijerph-16-03020-f002:**
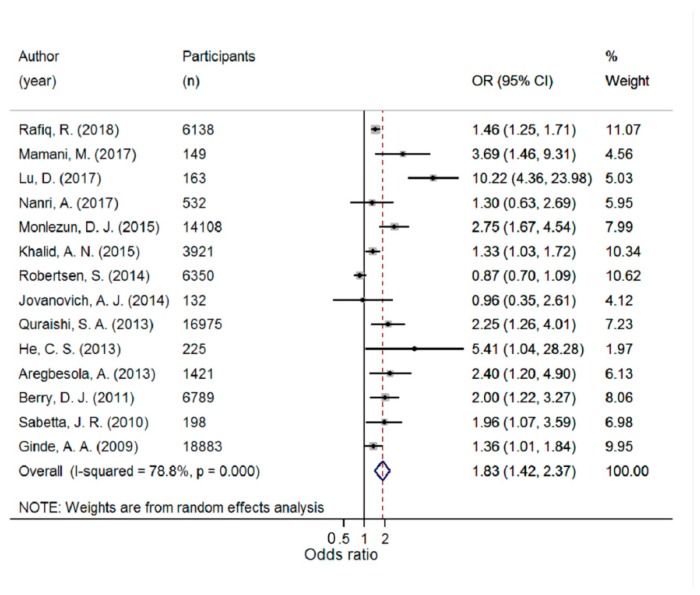
Forest plot displaying odds ratios (OR) and 95% confidence intervals (95% CI) for the association between 25(OH)D concentration and acute respiratory tract infection, comparing the lowest versus the highest 25(OH)D category.

**Figure 3 ijerph-16-03020-f003:**
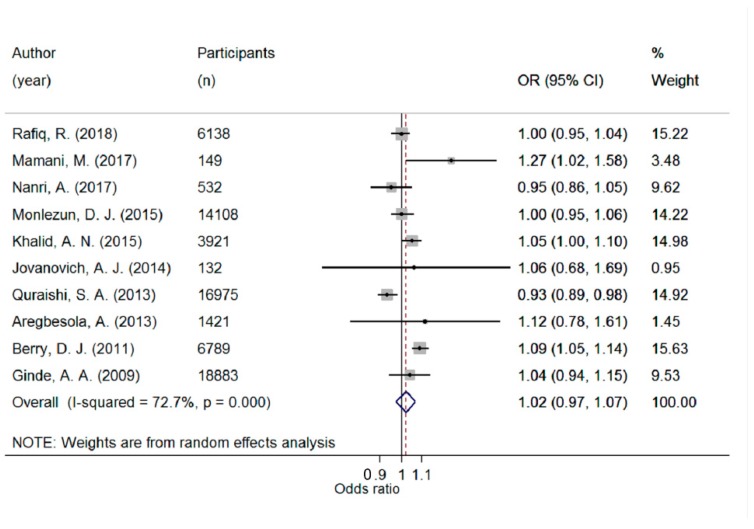
Forest plot displaying odds ratios (OR) and 95% confidence intervals (95% CI) of acute respiratory tract infection risk per 10 nmol/L decrease in 25(OH)D concentration.

**Figure 4 ijerph-16-03020-f004:**
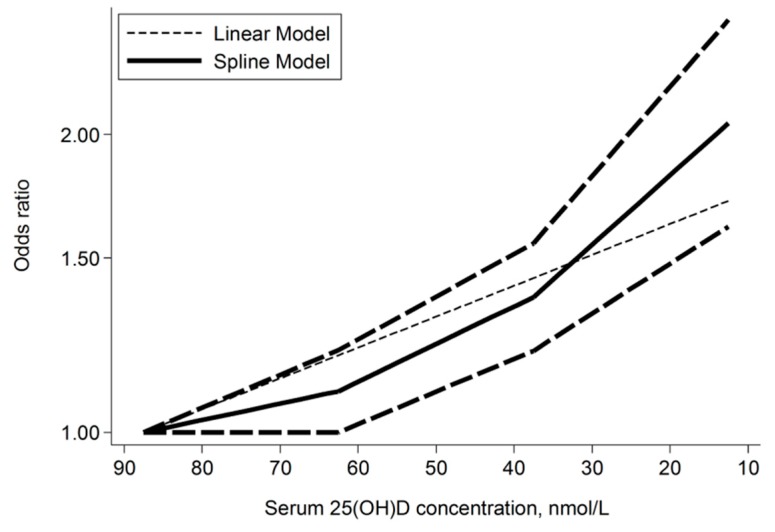
Dose-response relationship between serum 25(OH)D concentration and odds ratio of acute respiratory tract infection (*p* for non-linearity = 0.029). Lines with long dashes represent the upper and lower confidence interval for the fitted non-linear trend (solid line).

**Figure 5 ijerph-16-03020-f005:**
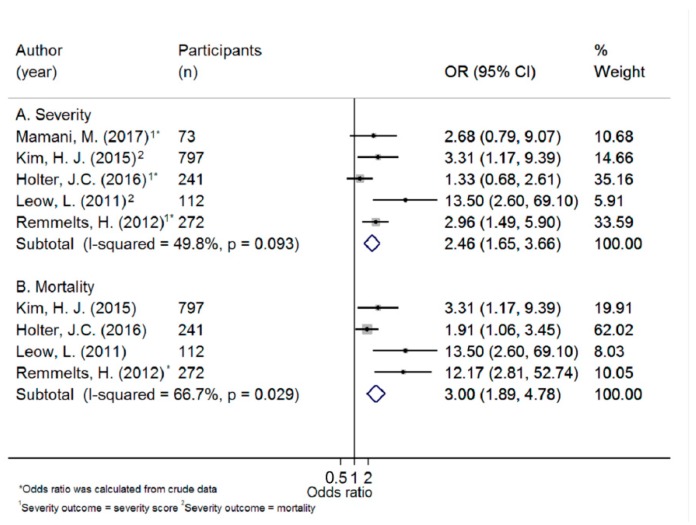
Forest plot displaying odds ratios (OR) and 95% confidence interval (95% CI) for the association between 25(OH)D concentration and (A) severe acute respiratory tract infection, and (B) mortality, comparing the lowest versus the highest 25(OH)D category.

**Figure 6 ijerph-16-03020-f006:**
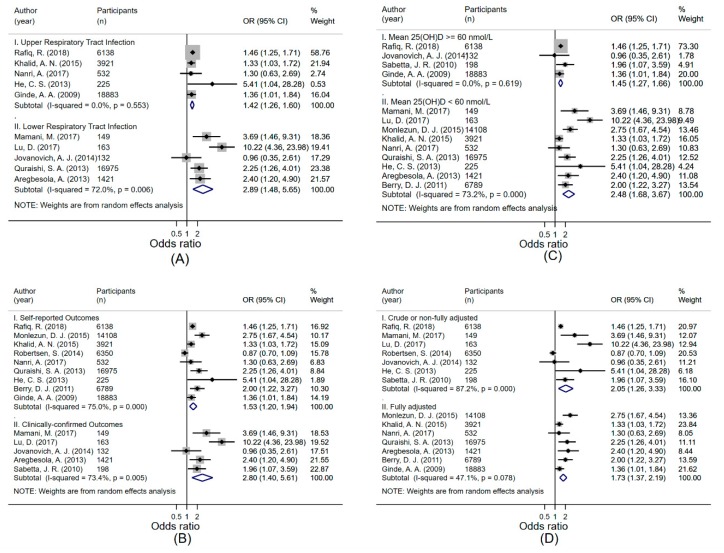
Forest plot displaying odds ratios (95% confidence intervals) for the association between 25(OH)D concentration and (**A**) (i) upper respiratory tract infection, and (ii) lower respiratory tract infection; (**B**) (i) self-reported, and (ii) clinically confirmed acute respiratory tract infection; (**C**) acute respiratory tract infection in studies with (i) mean 25(OH)D concentration ≥ 60 nmol/L, and (ii) mean 25(OH)D concentration < 60 nmol/L; and (**D**) acute respiratory tract infection in studies with (i) crude or non-fully adjusted effect estimate; and (ii) fully adjusted effect estimate; comparing the lowest versus the highest 25(OH)D category.

**Figure 7 ijerph-16-03020-f007:**
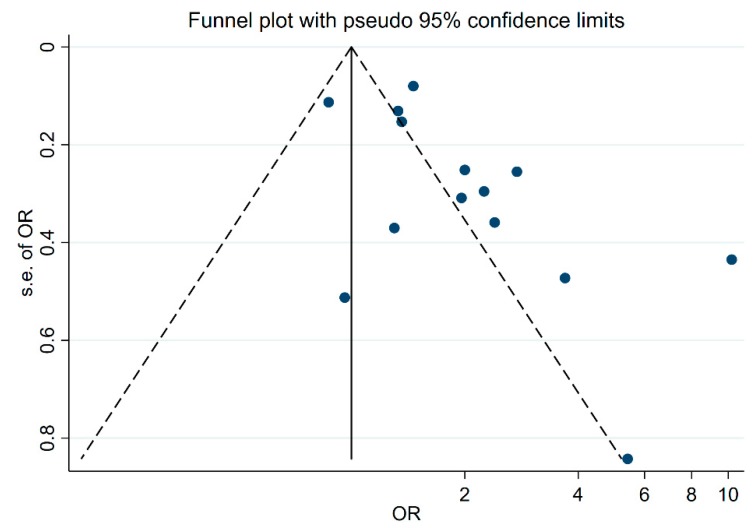
The funnel plot to check for publication bias.

**Table 1 ijerph-16-03020-t001:** Characteristics of studies reporting the association between risk of acute respiratory tract infection and 25(OH)D concentration.

First Author (Publication Year)	Year of Study	Follow-Up Time	Outcome	Participants	Age (Mean ± SD)/Sex (% Female)	Sample Size	Outcome Measurement	Mean (SD)/Median (Q1–Q3) 25(OH)D (nmol/L)	Lowest 25(OH)D Category (nmol/L)	Highest 25(OH)D Category (nmol/L)	25(OH)D Measurement Method
**Case-Control Studies**											
Jovanovich, A. J. (2014) [32]	2008–2010		Pneumonia	Cases: patients with CAP Controls: patients without CAP	Age: 60 ± 17 Sex: 71% F	132	Laboratory or x-ray confirmed	Cases: 70.1 (62.2–79.6) Controls: 79.3 (71.1–88.1)	50	≥50	INCSTAR RIA
Nanri, A. (2017) [35]	2011		Influenza	Cases: employees with influenza Controls: employees without influenza	Age: 38 ± 12 Sex: 17% F	532	Self-reported (past 6 months)	Cases: 56.1 (12.8) Controls: 55.9 (13.0)	50	≥75	CBP assay
Mamani, M. (2017) [45]	NA		Pneumonia	Cases: patients with CAP Controls: patients’ companions	Age: >18 Sex: 30% F	149	Laboratory or x-ray confirmed	Cases: 54.7 (61.9) Controls: 48.1 (27.8)	25	>50	Diarosin CLIA
**Cross-Sectional Studies**											
Ginde, A.A. (2009) [29]	1988–1994		URTI	NHANES III (1988–1994)	Age: ≥12 Sex: 53% F	18,883	Self-reported (past few days)	72.2 (52.3–92.1)	25	≥75	Diasorin RIA
Quraishi, S.A. (2013) [37]	1988–1994		Pneumonia	NHANES III (1988–1994)	Age: ≥17 Sex: 53% F	16,975	Self-reported (past 12 months)	59.8 (44.8–79.7)	25	≥75	Diasorin RIA
Khalid, A.N. (2015) [33]	2001–2006		Acute rhinosinusitis	NHANES 2001–2006	Age: ≥17 Sex: 51% F	3921	Self-reported (past 24 h)	54.8 (39.8–69.7)	50	≥50	Diasorin RIA
Monlezun, D.J. (2015) [34]	2001–2006		ARTI	NHANES 2001–2006	Age: ≥17 Sex: 51% F	14,108	Self-reported (past 1 month)	52.3 (37.4–67.2)	25	≥75	Diasorin RIA
Berry, D.J. (2011) [31]	2002–2004		ARTI	Birth cohort born 1958	Age: 45 Sex: 50% F	6789	Self-reported (past 3 weeks)	52.2	25	≥100	IDS OCTEIA
Robertsen, S. (2014) [46]	2007–2008		ARTI	Tromsø population-based study	Age: ≥40 Sex: 57% F	6350	Self-reported (past 7 days)		NA	NA	Roche CLIA
Rafiq, R. (2018) [38]	2008–2012		Common cold	NEO study, BMI ≥ 27 kg/m^2^	Age: 45–65 Sex: 56% F	6138	Self-reported (past 1 month)	71.3	50	≥75	Diasorin RIA, IDS CLIA, Roche CLIA
Lu, D. (2017) [13]	2011–2012		Pneumonia	Hospitalised patients	Age: 60–94 Sex: 31% F	163	Clinically diagnosed	30.0 (11.2)	25	≥25	IDS ELISA
Scullion, L. (2018) [47]	NA		ARTI	Elite rugby players and rowers	Age: 23 ± 3 Sex: 25% F	54	Self-reported (past 6 months)	Summer 108.9 (102.8–115.4) Winter 86.8 (81.8–92.1)	NA	NA	Crystal Chem enzymatic assay
**Prospective Cohort Studies**											
Aregbesola, A. (2013) [25]	1998–2011	10 years	Pneumonia	KIHD study: middle age and aging people	Age: 53–73 Sex: 49% F	1421	Clinically diagnosed	43.5 (17.8)	tertile 1: 8.9–33.8	tertile 3: 50.8–112.8	HPLC
Porojnicu, A. C. (2012) [36]	2007	winter season	ARTI	Medical employees from a hospital	Age: 20–57 Sex: 94% F	105	Laboratory or x-ray confirmed		NA	NA	HPLC
Sabetta, J.R. (2010) [26]	2009–2010	5 months	Viral ARTI	Healthy adults	Age: 20–88 Sex: 57% F	198	Clinically diagnosed Laboratory confirmed	71.0 (2.0)	95	≥95	Diasorin CLIA
He, C-S. (2013) [15]	2011	4 months	URTI	Young athletes	Age: 18–40 Sex: 30% F	225	Self-reported (4 month diaries)	53.0 (40.0–66.0)	12–30	>120	HPLC

Abbreviations: ARTI = acute respiratory tract infection; CAP = community acquired pneumonia; CBP = competitive protein binding; KIHD = Kuopio Ischemic Heart Disease Risk Factor; IDS=Immunodiagnostic Systems; RIA=radioimmunoassay; CLIA=chemiluminescence immunoassay; HPLC = high performance liquid chromatography; LRTI = lower respiratory tract infection; MS = mass spectrometry; NA = not available; NEO = Netherlands Epidemiology of Obesity study; NHANES = National Health and Nutrition Examination Survey; URTI = upper respiratory tract infection.

**Table 2 ijerph-16-03020-t002:** Characteristics of studies reporting the association between severity of ARTI and 25(OH)D.

First Author (Publication Year)	Year of Study	Follow-Up Time	Outcome	Participants	Age (Mean ± SD)/Sex (% Female)	Sample Size	Severity Measurement	Mean (SD)/Median (IQR) 25(OH)D (nmol/L)	Lowest 25(OH)D Category (nmol/L)	Highest 25(OH)D Category (nmol/L)	25(OH)D Measure ment Method
**Cross-Sectional Studies**											
Mamani, M. (2017) [45]	NA		Pneumonia severity	Hospitalised patients with CAP	Age: 68 ± 10 Sex: 29% F	73	CURB-65 > 2 ICU admission Death	Severe: 52.8 (77.5) Non-severe: 56.5 (48.0) Yes: 59.8 (88.4) No: 52.8 (47.6) Yes: 64.6 (91.8) No: 53.0 (55.4)	<25	≥75	Diasorin CLIA
Pletz, M. W. (2014) [14]	2002–2008		Pneumonia severity	Participants with pneumonia	Age: ≥18 Sex: 43% F	300	Hospitalisation	Severe: 32.0 (19.5) Non-severe: 40.5 (25.0)	NA	NA	Diasorin CLIA
Robertsen, S. (2014) [46]	2007–2008		ARTI	Tromsø population-based study	Age: ≥40 Sex: % F	791	Duration of the illness		NA	NA	Roche CLIA
Lu, D. (2017) [13]	2011		Pneumonia severity	Patients with pneumonia	Age: 60–94 Sex: 31% F	49	Duration of hospitalisation		<25	≥25	IDS ELISA
Kim, H.J. (2015) [41]	2012–2014		Pneumonia severity	Hospitalised patients with CAP	Age: 18–96 Sex: 34% F	797	28-day all-cause mortality Need for mechanical ventilator		<50	≥50	CLIA
Brance, M. (2018) [39]	2015–2016		Pneumonia severity	Hospitalised patients with CAP	Age: >18 Sex: 59% F	167	CURB-65 ≥ 2	Severe: 29.0 (18.3) Non-severe: 29.8 (18.8)	<25	>50	Siemens CLIA
Yaghoobi, M.H. (2019) [44]	2015		Ventilator-associated pneumonia	Hospitalised patients with ventilator-associated pneumonia	Age: 18–82 Sex: 37% F	84	Mortality in 28 days Blood culture Duration of ventilation SOFA score	Yes: 61.5 (23.7) No: 61.9 (20.8) Positive: 53.4 (12.3) Negative: 62.7 (22.3)	<75	≥75	NA
Scullion, L. (2018) [47]	NA		ARTI	Elite rugby players and rowers	Age: 23 ± 3 Sex: 25%F	53	Duration of the illness	Summer: 4.8 (3.0) Winter: 6.9 (4.3)	NA	NA	Crystal Chem enzymatic assay
**Prospective Cohort Studies**											
Laaksi, I. (2007) [16]	2002	6 months	ARTI severity	Young military men	Age: ≥18 Sex: 0% F	652	Number of days absence from duty due to ARTI		<40	≥40	IDS OCTEIA
Remmelts, H.F. (2012) [43]	2007–2010	30 days	Pneumonia severity	Hospitalised patients with CAP	Age: ≥18 Sex: 44% F	272	ICU admission 30-day mortality PSI > 3	ICU: 34.9 (23.8–46.3) No ICU: 48.3 (30.8–68.4) Yes: 25.8 (19.8–40.1) No: 48.8 (32.4–68.9)	<50	>75	Diasorin CLIA
Holter, J.C. (2016) [40]	2008–2014	6 years	Pneumonia severity	Hospitalised patients with CAP	Age: ≥18 Sex: 49% F	241	CURB-65 ≥ 3 ICU admission Long-term all-cause mortality		<30	≥50	Siemens CLIA
Leow, L. (2011) [42]	2008	4 months	Pneumonia severity	Hospitalised patients with CAP	Age: 16–97 Sex: NA	112	30-day mortality		<30	>50	Roche CLIA
Sabetta, J.R. (2010) [26]	2009–2010	5 months	Viral ARTI severity	Healthy adults	Age: 20–88 Sex: 57% F	198	Duration of the illness		<95	≥95	Diasorin CLIA
He, C-S. (2013) [15]	2011	4 months	URTI severity	Young athletes	Age: 18–40 Sex: 31% F	103	Duration of the illness SSC		12–30	>120	HPLC

Abbreviations: ARTI = acute respiratory tract infection; CAP = community acquired pneumonia; CURB-65 = confusion, uremia, respiratory rate, low blood pressure, age ≥ 65 years; ICU = intensive care unit; LC = liquid chromatography; LRTI = lower respiratory tract infection; MS = mass spectrometry; NA = not available; PSI = pulmonary severity score; SSC = symptom severity score; URTI = upper respiratory tract infection.

**Table 3 ijerph-16-03020-t003:** Studies reporting duration of illness according to 25(OH)D concentration.

First Author (Published Year)	Duration of Illness Mean (SD)/Median (IQR) (Days)		*p*-Value
	Lowest 25(OH)D Category	Highest 25(OH)D Category	
Mamani, M. (2017) ^1^	11.03 (7.5)	9.47 (6.1)	
Lu, D. (2017) ^1^	26.2 (15.6)	15.5 (11.1)	0.014
He, C-S. (2013) ^2^	13 (10–17)	5 (5–7)	≤0.05
Sabetta, J.R. (2010) ^2^	6 (2–8)	6 (2–27)	
Laaksi, I. (2007) ^3^	4 (2–6)	2 (0–4)	0.004
Kim, H.J. (2015) ^1^	12.5 (15.4)	10.3 (11.0)	0.570
Robertsen, S. (2014) ^2^	14	13	
Yaghoobi, M.H. (2019) ^4^	13.4 (6.1)	13.7 (9.8)	0.880
Holter, J.C. (2016) ^2^	4 (2–6)	5 (3–10)	
Scullion, L. (2018) ^2,a^	6.9 (4.3)	4.8 (3.0)	0.044

Abbreviations: IQR = interquartile range; SD = standard deviation. ^1^ Duration of hospitalisation; ^2^ Duration of symptoms; ^3^ Number of days of absence from duty due to acute respiratory tract infection; ^4^ Duration of mechanical ventilation. ^a^ The study reported duration of the symptoms, comparing winter and summer; the mean 25(OH)D for winter was 86.8 nmol/L and for summer was 108.9 nmol/L.

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
