# Peer review of "Acute Respiratory Tract Infection and 25-Hydroxyvitamin D Concentration: A Systematic Review and Meta-Analysis"

_ijerph, 2019, doi:10.3390/ijerph16173020_

Round 1
Reviewer 1 Report
Congratulations on a very well written and concise paper that even a person less familiar with systematic reviews and Meta-analyses found easier to follow than most.
As a reviewer my comments are minor and should be taken as suggestions to encourage future observational studies to minimize confounding elements.
Table 1 and 2: It would be helpful to include information describing the analytical method used to measure serum 25(OH)D concentration since the variability in outcome due to the analytic method used and the absence of standardization represent a significant source of confounding as described by: Sempos CT et al. Scand J. Clin & Lab Inv. 2012; 72 (Suppl 243): 32-40. and Sempos CT et al. Vitamin D assays and the definition of hypovitaminosis D: Results from the first International Conference on Controversies in Vitamin D. Br J Clin Pharmacol. 2018; 84(10): 2194-2207, or Giustina A et al. Controversies in Vitamin D: Summary Statement from an International Conference. J Clin Endocrinol Metab. 2019; 104(2):234-240. Page 13, Lines 25-26: suggest including: ...because of variability across studies with respect to...;"variability in assay methods used to measure serum 25(OH)D" and the measure used to estimate effect.Author Response
Please see the attachment.

Reviewer 2 Report
The review “Acute respiratory tract infection and 25-hydroxyvitamin D concentration: a systematic review and meta-analysis”, has as its main objective to establish the relationship between 25(OH)D serum concentration and ARTI, is very updated and with proper methodology.
The following corrections/suggestions are indicated:
- Specify the search articles period in databases;
- Clarify the cut off points for 25(OH)D and the analysis methodologies, in the different statistical relationships established;
- Correct formatting and spelling errors in table 1;
- Item 5 of the inclusion criteria are only healthy individuals, adults or adolescents. No data were reported involving children?
The study is scientifically relevant in establishing the relationship between serum 25(OH)D concentration and ARTI severity; however, reference values for 25(OH)D supplementation are not established. As well as there is no discussion between the 25(OH)D serum concentration and the risk and severity of ARTI by age group, and this condition affects children to the elderly.
